# Idiopathic intracranial hypertension in patients with anemia: A retrospective observational study

**Zhonghua Ma¤, Hanqiu Jiang¤, Chao Meng¤, Shilei Cui¤, Jingting Peng¤, Jiawei Wang¤***

Department of Neurology, Beijing Tongren Hospital, Capital Medical University, Beijing, China

¤ Current address: Department of Neurology, Beijing Tongren Hospital, Xicheng District, Beijing, China
* wangjwcq@163.com

**Data Availability Statement:** All relevant data are within the manuscript and its Supporting Information files.

**Funding:** Funded by "Beijing Tongren Hospital, Capital Medical University, Key Medical

## Abstract

Idiopathic intracranial hypertension (IIH) mostly affects obese women in childbearing age, leading to frustrating headache and permanent visual impairment. The exact etiology of this condition is poorly understood, and the population at risk and clinical presentation seems to be homogeneous. However, little attention has been paid to the clinical features of IIH patients with anemia. We herein performed a retrospective observational study by using the data of patients with presumed IIH who were referred to the neurology department of Beijing Tongren Hospital from January 2014 to August 2019 to describe the clinical features and radiological findings in patients with IIH and anemia, and compared these with those without anemia. The patients were divided into two groups based on the presence of anemic diseases. Clinical data including demographic characteristics, clinical features, past medical history, laboratory and neuroradiological findings, diagnoses, treatments and prognosis of these patients were reviewed and compared in both the groups. A total of 153 patients with IIH were enrolled, which included 22 cases with anemia (mean age, 33.23±9.68 years; 19 [86.36%] female) and 131 cases without anemia (mean age 37.11±11.56 years; 97 [74.05%] female). In the anemia group, 19/22 cases had iron deficiency anemia and 3/22 had renal anemia. Compared with patients in the non-anemia group, IIH patients with anemia had a shorter disease course, and tended to present pulsatile tinnitus and transverse sinus stenosis (TSS), faster and better prognosis after treatments for correcting anemia and reducing intracranial pressure. Our findings highlighted the importance of obtaining full blood counts in IIH patients with subacute onset, and provided appropriate and prompt treatments if proven anemic in order to bring better outcomes.

## Introduction

Idiopathic intracranial hypertension (IIH) is a complex condition characterized by headache, blurred vision and papilledema with persistently high intracranial pressure (ICP), normal cerebrospinal fluid (CSF) composition and normal neuroimaging results. The precise etiology of this condition still remains unclear, but diagnostic accuracy is considered essential due to the

Development Plan, TRYY-KYJJ-2017-054". The funder had no role in study design, data collection and analysis, decision to publish, or preparation of the manuscript.

**Competing interests:** The authors have declared that no competing interests exist.

risk of devastating vision loss. The risk factors of IIH include obesity, anemia, and use of steroids [1–3]. Currently, there is limited literature focusing on the manifestations of IIH in patients with anemia. Hence, this study aimed to demonstrate the clinical characteristics and prognosis of IIH patients with anemia.

## Material and methods

### Cohort

The records, charts and examinations from all patients with presumed IIH referred to the neurology department of Beijing Tongren Hospital (Beijing, China) from January 2014 to August 2019 were retrospectively reviewed. The patients with IIH were diagnosed based on the modified Dandy' criteria [4], which include papilledema; normal neurologic examination except for cranial nerve abnormalities; neuroimaging: normal brain parenchyma without evidence of hydrocephalus, mass, or structural lesion and no abnormal meningeal enhancement on MRI; normal CSF composition; elevated lumbar puncture opening pressure (>25cm in adults) during a properly performed lumbar puncture. The exclusion criteria were as follows: patients 1) with masses in the central nervous system (CNS) or other primary causes that increased ICP, 2) who underwent initial treatment in other hospitals before attending to Beijing Tongren Hospital, 3) with confounding neurological comorbidities, and 4) with unavailable neurological imaging findings, or clinical details, or signed consent. All patients were re-identified by two authors and a radiologist before enrolling into our registry. Most of the patients in this study were seen after diagnosis of IIH at about 3 and 6 months. A trained doctor followed-up the patients through telephone, and monitored treatment compliance.

As shown in Fig 1, 40 patients were excluded: 16 due to other potential causes of visual loss than IIH, 21 missed follow-up visits within the six months after diagnosis and 3 underwent treatments initiated by other hospitals. Finally, 153 patients were considered eligible and included in this study. These patients were divided into two groups based on the presence of anemic diseases. The clinical features were described and compared between the two groups.

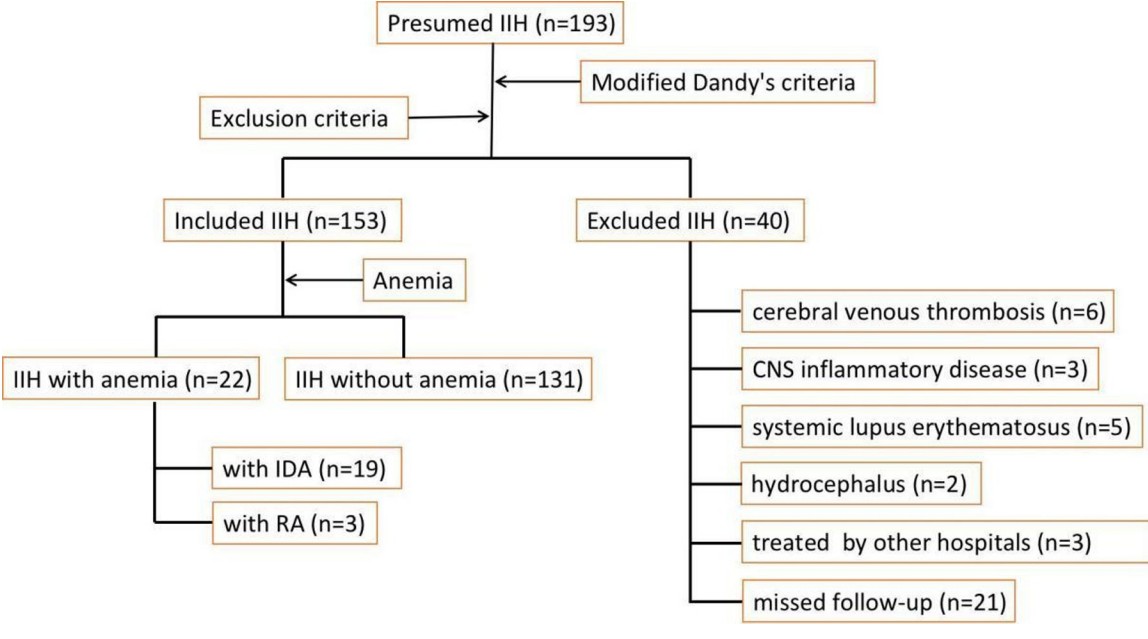

**Fig 1. Flow chart of cases' enrollment.**

The diagnostic criteria of anemia in Chinese population include: hemoglobin (Hb) <120g/L, hematocrit (Hct) <0.42 for adult male; Hb <110 g/L, Hct<0.37 for adult female; the severity of anemia is graded as mild with Hb ≥90 g/L, moderate with 60g/L≤Hb≤89g/L, severe anemia with 30g/L≤Hb≤59g/L, and severe anemia with Hb<30g/L. Iron deficiency anemia (IDA) is defined as hypochromic microcyticanemia with ferritin <15 μg/L. Renal anemia (RA), which is also known as anemia of chronic kidney disease, is defined as anemia with glomerular filtration rate (GFR) of <60 ml/min/1.73m$^2$, and no other cause, e.g. blood loss, folic acid or vitamin B12 deficiency.

All patients provided signed the written informed consent form to include in this clinical research, and this study was approved by the Ethics Committee of the Capital Medical University. All enrolled participants signed the consent form for research and publication of their clinical data and questionnaire responses.

## Data collection

The improvement in patient's condition and study patients before and after medical treatment were observed. Thus, we retrospectively reviewed at four study points: baseline visit (VST 0), follow-up visit 1 at discharge (20days, VST 1), follow-up visit 2 at 2–4 months (VST 2) and follow-up visit 3 at 5–8 months (VST 3).

At baseline, the demographic characteristics (i.e., age at onset of disease, sex, height and weight), clinical data including date of onset of the disease and date of the first presentation (used to calculate the disease course), clinical manifestations, past medical history, lumbar

CSF pressure, laboratory (including CSF analysis) and neuroradiological examination (such as MRI/MRV/CTV/DSA) findings, diagnoses, treatments of these patients were collected and reviewed. The actual body weight was categorized into three types based on the classification of overweight and obesity of Chinese adults: 18.5≤ BMI <24, normal; 24≤BMI<28, overweight; and BMI≥28, obese. The "course of disease" was defined as the time from onset to diagnosis.

The changes in best corrected visual acuity (BCVA), headache impact, papilledema grade, automated perimetry, CSF open pressures (OP) from VST 0 to VST 3 were reviewed and analyzed. All patients treated by surgical measures are not included in the follow-up statistical analyses.

## Visual examinations

*BCVA* was tested according to the international visual chart. Grading of visual impairment was done based on the visual acuity: severe, BCVA < 0.1; moderate, 0.1≤ BCVA < 0.5; and mild, BCVA ≥ 0.5.

*Papilledema* was assessed by fundus photochromy in all patients. The eye with higher degree of papilledema was chosen as the study eye at baseline. If both eyes had equal degree of papilledema, the right eye was chosen. The papilledema grade (modified Frisérisfied, MFS) [5,6] was independently evaluated by two neuro-ophthalmologists; and the values range from 0 (normal) to 5 (severe papilledema). In case of any disagreements in the classification, the fundus photochromy was adjudicated by reaching a consensus based on the group.

*Visual field (VF) examination* was done for patients by using the Humphrey Field (HF) Analyzer (Model 750; Carl Zeiss Meditec Inc, Dublin, CA, USA). Unreliable autoperimetry (false-positive errors and false-negative errors >20% or false-positive errors >20%) were excluded. The eye with worse mean deviation (MD) at baseline was used as the study eye. Two neuro-ophthalmologists independently evaluated these VFs according to the perimetry MD grading scale established by Shah et al. [7]: Grade 0- normal visual field; Grade 1-MD less than

4.0 with a visual field defect; Grade2—MD 4.0 to 11.9 with significant visual field defect; Grade 3-MD 12.0 to 19.9; and Grade 4-MD greater than 20.0.

## Headache

IIH related headache was defined according to the International Classification of Headache Disorders version 3 beta (ICHD-3β) diagnostic criteria for headache attributed to IIH [8]. The visual analogue scale (VAS) of headache (0–10 scale) and the 6-item Headache Impact Test (HIT-6) were used for assessing the severity and impact of headache in patients from VST 0 to VST 3.

## Laboratory tests

At baseline, conventional routine blood tests, biochemical tests and coagulation examinations were performed. If anemia was found in the patient, then anemia-related tests were performed to clarify the causes. Serum aquaporin-4 antibody (anti AQP4-IgG), double-stranded DNA (dsDNA) antibody, antinuclear antibody (ANA), extractable nuclear antigen (ENA) antibodies, C-reactive protein (CRP), antineutrophil cytoplasmic antibodies (ANCA), and mitochondrial DNA (mt-DNA) mutation tests were observed. Intracranial hypertension (elevated CSF OP ≥25cmH$_2$O) was confirmed by open lumber puncture (LP) with the patient lying in the lateral decubitus position. CSF analysis was performed for all patients to exclude any infection/inflammatory events. The results of Hb tests in the anemic group before and after treatment were collected from VST 0 to VST 3.

## Neurological imaging

All patients underwent gadolinium (Gd)-enhanced brain and orbital MRI, including T1WI, T2WI sequences with fat suppression to rule out optic neuropathy. The axial, coronal and sagittal MR imaging were performed on a Discovery MR750 3.0-T MR imaging system (GE Healthcare, Milwaukee, WI) with an 8-channel head coil. A standard dose (0.1 mmol/kg) of gadolinium based contrast agent was intravenously administered at 2.0 cc/second. Each MR image was reinterpreted by a neuroradiologist who was masked to the affected side. Bilateral distension of the perioptic subarachnoid space (DPSS), and empty sella turcica were assessed with coronal and sagittal T2WI, respectively; while the venous sinus thrombosis, narrowing related to IIH, or congenital narrowing of the venous sinuses were evaluated on CTV/DSA. The images were then used for independently assessing the clinical data.

## Treatments

Based on weight control, all patients underwent treatment with acetazolamide (25mg/tablet) with an initial dose of 4 tablets daily, followed by increased dosage of 2 tablets every 5 days, and up to a dose of 200-300mg daily for those receiving acetazolamide. 20% mannitol (125-250ml, intravenous infusion, 3–4 times/day) was also used for patients with obvious headache. Based on the condition of continuous worsening of symptoms in patients due to conventional medical therapy or very poor visual acuity at presentation, the patients were offered LP decompression while waiting for a surgical intervention. For patients with IDA, ferrous succinate tablets at a dose of 300-400mg/day were given. Patients with RA were treated with hypodermic erythropoietin by nephrologists.

Finally, 21/22 patients in anemia group received routine treatments including oral acetazolamide, intravenous infusion of mannitol, decompression by LP monotherapy, and anemia-correcting therapy. Only 1/22 anemic patient received surgical treatment with lumbo-

peritoneal shunt at baseline due to bad vision. In contrast, 117/131 cases received routine treatments, and 14/131 underwent surgical procedures (8 at baseline and 6 after VST 2 for worsening eyesight) in the non-anemic group.

## Statistical analysis

Statistical analysis was performed using SPSS software version 22.0. Categorical variables were reported as numbers and percentage. Continuous variables were presented as means and standard deviation. Independent sample *t* tests and Mann-Whitney *U* tests were performed to compare continuous variables. Pearson chi-square test and Fisher's exact test were used to compare the frequencies. *P* values were 2-tailed, and statistical significance was set at 0.05.

## Results

### Demographic characteristics

A retrospective non-interventional clinical observation of 153 patients with IIH was done. Of these adult patients, 22 had anemia (mean age at disease onset, 33.23±9.68 years; 19 [86.36%] women) and 131 had no anemia (mean age at disease onset, 37.11±11.56 years; 97 [74.05%] women). The demographic characteristics of each group are presented in Table 1. Both groups included obese and overweight patients (8 obese [36.36%], 8 overweight [36.36%] in patients with anemia; 43 obese [32.82%], 62 overweight [47.33%] in patients without anemia). There was no significant difference in age, gender, and BMI between the two groups.

### Clinical presentation at baseline

As shown in Table 2, the median course of duration for patients with anemia (1 month) was significantly shorter than that for those without anemia (3 months) (Z = -3.985, *P* = 0.00). The predominant symptoms in both groups included vision loss (81.81% in group with anemia vs. 87.78% in group without anemia, *P* = 0.317), followed by headache (68.18% vs 51.91%, *P* = 0.154), transient visual obscuration (40.91% vs 51.91%, *P* = 0.340), pulsatile tinnitus (54.55% vs 29.01%, *P* = 0.018), and double vision (27.27% vs 13.74%. *P* = 0.118). IIH patients with anemia more likely developed pulsatile tinnitus than those without anemia (54.55% vs 30.53%, *P* = 0.018). Physical examinations, such as funduscopic examination revealed papilledema in different degrees in all patients, and pale optic discs were observed in 22 eyes; and sixth cranial nerve palsy was seen in 20 subjects. Other neurological physical examinations were found to be negative. The intraocular pressure was within the normal limits.

**Table 1. Demographic information of IIH patients in the two groups.**

|  | With anemia (n = 22) | Without anemia (n = 131) | P value |
|---|---|---|---|
| **Sex** |  |  |  |
| Female | 19 (86.36%) | 97 (74.05%) |  |
| Male | 3 (13.64%) | 34 (25.95%) | 0.212 |
| **Age(years)** | 33.23+9.68 | 37.11+11.56 | 0.269 |
| **BMI** |  |  |  |
| <24 | 6 (27.27%) | 26 (19.85%) | 0.408 |
| (24,28) | 8 (36.36%) | 62 (47.33%) | 0.339 |
| ≥28 | 8 (36.36%) | 43 (32.82%)0.408 | 0.745 |

**Table 2. Clinical and neuroradiological characteristics of IIH patients in the two groups at baseline.**

|  | With anemia (n = 22) | Without anemia (n = 131) | P value |
|---|---|---|---|
| **Course of disease (month)** | M = 1.00, Q = 1.43 | M = 3.00, Q = 6.00 | 0.00 |
| **Symptoms** |  |  |  |
| Vision loss | 18 (81.81%) | 115 (87.78%) | 0.317 |
| Head an neck pain | 15 (68.18%) | 68 (51.91%) | 0.154 |
| Transient obscure | 9 (40.91%) | 68 (51.91%) | 0.340 |
| Tinnitus | 12 (54.55%) | 38(29.01%) | 0.018 |
| Double vision | 6 (27.27%) | 18(13.74%) | 0.118 |
| **Abnormal image (overall)** | 16 (72.73%) | 97 (74.05%) | 0.896 |
| Empty sella | 10 (45.45%) | 65 (49.62%) | 0.718 |
| DPSS | 5 (22.73%) | 29 (22.14%) | 1.000 |
| TSS | 12 (54.55%) | 40 (30.53%) | 0.028 |
| **Misdiagnosis** | 7(31.82%) | 56 (42.75%) | 0.335 |

M, median; Q, quartile. DPSS, distension of the perioptic subarachnoid space. TSS, transverse sinus stenosis.

## Visual examinations

**BCVA.** At VST 0, 17 anemic cases with 34 eyes had decreased vision, in which 12/34 eyes (35.29%) with BCVA<0.1, 10/34 eyes (29.41%) with $0.1 \leq$ BCVA<0.5, and 12/34 eyes (35.29%) with BCVA$\geq$0.5. While in the non-anemic group, 101 cases with 193 eyes had bad eyesight, in which 48/193 eyes (24.87%) were with BCVA<0.1, 27/193 eyes (13.99%) were with $0.1 \leq$ BCVA<0.5, and 118/193 eyes (61.14%) were with BCVA$\geq$0.5. Statistical analysis revealed higher proportion of patients with $0.1 \leq$ BCVA<0.5 in anemia group ($P$ = 0.025), while higher proportion of patients with BCVA$\geq$0.5 those without anemia ($P$ = 0.005) as shown in Table 3. At VST 1, i.e., about 20 days after the treatment, the frequency of severe visual impairment (BCVA<0.1) in anemia group was significantly lower than that in non-anemia group (5.88% vs 21.76%, $P$ = 0.031, Table 2). The BCVAs were similar between both groups at VST 2. However, at VST 3, more patients in anemia group had better BCVA (above 0.5, 82.35% vs 65.28%, $P$ = 0.049).

**Papilledema.** Twenty one eyes in anemia group and 117 eyes in non-anemia group were under observation as shown in Table 4. A total of 15 eyes [71.43%] in anemia group and 69

**Table 3. Comparison of improvement of BCVA in the anemia group and non-anemia group.**

| BCVA | With anemia (n = 34) | Without anemia (n = 193) | P value |
|---|---|---|---|
| **VST 0**<0.1 | 12(35.29%) | 48 (24.87%) | 0.204 |
| (0.1,0.5) | 10 (29.41%) | 27 (13.99%) | 0.025 |
| $\geq$0.5 | 12 (35.29%) | 118 (61.14%) | 0.005 |
| **VST 1**<0.1 | 2 (5.88%) | 42(21.76%) | 0.031 |
| (0.1,0.5) | 8(23.53%) | 30(15.54%) | 0.250 |
| $\geq$0.5 | 24(70.59%) | 121(62.69%) | 0.377 |
| **VST 2**<0.1 | 2 (5.88%) | 34 (17.62%) | 0.124 |
| (0.1,0.5) | 6 (17.65%) | 40 (20.73%) | 0.681 |
| $\geq$0.5 | 26 (76.47%) | 119 (61.66%) | 0.097 |
| **VST 3** <0.1 | 2(5.88%) | 30(15.54%) | 0.183 |
| (0.1,0.5) | 4(11.76%) | 37 (19.17%) | 0.301 |
| $\geq$0.5 | 28(82.35%) | 126(65.28%) | 0.049 |

**Table 4. Comparison of follow-up information in anemia group and non-anemia group.**

| | MFS grade | | | VF Defect Grade | | | HIT-6 Score | | | CSF OP(cmH2O) | | |
|---|---|---|---|---|---|---|---|---|---|---|---|---|
| | Anemic (n = 21) | Non-Anemic (n = 117) | P value | Anemic (n = 12) | Non-Anemic (n = 62) | P value | Anemic (n = 13) | Non-Anemic (n = 56) | P value | Anemic (n = 21) | Non-Anemic (n = 117) | P value |
| VST0 | 3.91(0.97) | 3.47(1.10) | 0.082 | 3.00(1.20) | 2.17(1.27) | 0.046 | 58.38 (5.19) | 57.46(3.34) | 0.427 | 38.17 (5.39) | 34.78(5.89) | 0.015 |
| VST1 | 3.09(0.92) | 3.13(0.88) | 0.849 | 2.75(1.29) | 1.81(1.31) | 0.034 | 52.84 ±3.53 | 54.05(3.75) | 0.294 | 32.21 (5.25) | 32.84(5.26) | 0.616 |
| VST2 | 2.05(0.84) | 1.93(0.68) | 0.484 | | | | 45.46 (4.14) | 43.96(2.67) | 0.324 | 24.24 (4.75) | 23.48(5.13) | 0.527 |
| VST3 | 0.86(0.71) | 0.76(0.61) | 0.454 | 1.25(1.22) | 0.82(0.96) | 0.269 | 41.69 (3.50) | 40.02(2.80) | 0.068 | 18.45 (3.42) | 17.89(2.63) | 0.395 |
| Improvement (VST1-VST0) | -0.82 (0.59) | -0.34(0.49) | 0.000 | -0.25 (0.45) | -0.36(0.57) | 0.436 | -5.54 (2.96) | -3.41(4.06) | 0.08 | -5.95 (2.18) | -1.41(1.47) | 0 |
| Improvement (VST2-VST0) | -1.86 (0.56) | -1.54(0.92) | 0.115 | | | | -12.92 (5.65) | -13.50(4.12) | 0.601 | -13.93 (3.74) | -10.77(3.60) | 0 |
| Improvement (VST3-VST0) | -3.05 (1.00) | -2.72(0.94) | 0.135 | -1.75 (0.75) | -1.35(0.73) | 0.112 | -16.69 (5.69) | -17.45(3.04) | 0.506 | -19.71 (4.62) | -16.22(5.33) | 0.005 |

Values are presented as means (SD) unless otherwise stated.

eyes [58.97%] in non-anemia group presented severe papilledema with grade 4–5 on MFS. Improvement in the grade of papilledema was observed in both groups. In anemia group, the mean MFS was dropped from 3.91 (0.97) at VST 0 to 0.86 (0.71) at VST 3, and 7 eyes [33.33%] returned to normal state at VST 3. In non-anemia group, the mean MFS was improved from 3.47(1.10) at VST 0 to 0.76(0.61) at VST 3, and papilledema of 44 eyes [37.61%] was completely resolved. The mean MFS was similar in both the groups at each follow-up visit. However, at VST 1, eyes in anemia group showed a significantly larger improvement (VST 1—VST 0) when compared to eyes in non-anemia group [-0.82(0.59) vs. -0.34(0.49), P<0.01]. While the improvement (VST 2—VST 0) and the improvement (VST3—VST 0) in mean MFS showed no significant differences between the two groups.

**Visual field.** There are only 74 participants (12 anemic, and 62 non-anemic) had HF perimetry results that met the inclusion criteria mentioned above. Visual field (VF) defect grade ranged from 0 to 4 throughout the study period. The improvements observed in VF in both groups were presented in Table 4. From VST 0 to VST 3, the mean VF defect grade in the anemia group was dropped from 3.00 (1.20) to 1.25 (1.22), and the mean grade in non-anemia group was dropped from 2.17 (1.27) to 0.82 (0.96). At VST 0 and VST 1, the mean VF defect grades of the anemia group were significantly higher than that of the non-anemic group [3.00 (1.20) vs. 2.17(1.27), P = 0.046; 2.75(1.29) vs. 1.81(1.31), P = 0.034].

However, the mean VF defect grades were similar between both groups at VST 3. There was no significant difference in the improvement degrees between the two groups at VST 1 and VST 3. The follow-up data of VST2 were not listed here, and for 36 participants (5 anemic and 21 non-anemic) who underwent visual field tests in other hospitals at VST 2, the results were not comparable with other follow-up visual field tests due to differences in the equipment used.

**Headache.** A total of 69 patients presented symptoms of headache at baseline (13 anemic and 56 non-anemic). Evaluation by VAS and HIT-6 indicated that most of these cases had moderate and substantial headaches. The mean headache severity at VST 0 was assessed based on a 0–10 rating scale (VAS), and was similar in both groups [6.7 (1.6) vs. 6.5 (1.7), P = 0.87].

There was no significant difference in HIT-6 score between the two groups of patients from VST 0 to VST 3 (Table 4). The mean HIT-6 score of patients in anemia group was decreased from 58.38(5.19) at VST 0 to 41.69 (3.50) at VST 3, and 10/13 patients got resolved completely at VST 3. While in non-anemic group, the mean HIT-6 score was decreased from 57.46 (3.34) at VST 0 to 40.02 (2.80) at VST 3, and 16/56 participants had complete relief at VST 3. There was no significant difference in the improvements of mean HIT-6 score between the two groups from VST 1 to VST 3.

## Laboratory tests

The results of serial serum immunological antibody tests were shown to be negative. Twenty two anemic cases had mean Hb of 97.45(9.12)g/L. Of these, 16/22 had mild anemia and 6/22 had moderate anemia (mean Hb 83.25 (5.69) g/L); 19/22 had IDA and 3/22 had RA. At the time of discharge (VST 1), the Hb of 5 patients had returned to normal, and the mean Hb in the remaining 17 was increased to 102.53 (6.34)g/L. At VST2, only 6 people had mild anemia (Hb 114, 106, 105, 101, 98, 96 and 99g/L). At VST3, the Hb of all patients has returned to normal.

## Lumbar puncture

CSF analyses at baseline, including routine tests, biochemical composition, oligoclonal bands (OB), myelin basic protein (MBP), intrathecal IgG synthesis rate, were in normal range. Reduction in ICP occurred in both groups. As shown in Table 4, the mean CSF OP of the anemia group was decreased from 38.17 (5.39) cm $H_2O$ at VST 0 to 18.45 (3.42)cm $H_2O$ at VST 3, but there are 6 patients [28.57%] with abnormal OP (ranging from 21 to 27 cmH2O) at VST 3. While in non-anemic group, the mean CSF OP was decreased from 34.78 (5.89) cm $H_2O$ at VST 0 to 17.89 (2.63) cm $H_2O$ at VST 3, but 27 participants [23.07%] had elevated CSF pressures of >20 cm $H_2O$ at VST 3 (ranging from 20 to 26 cm $H_2O$). At baseline, the average CSF OP of patients in the anemia group was significantly higher than in the non-anemic group [38.17 (5.39) vs. 34.78 (5.89) cmH$_2$O, $P<0.01$]. However, at VST1, VST2 and VST3, the mean CSF Ops showed no significant differences between the two groups. At discharge (VST 1), the CSF OP reduction in patients with anemia was significantly greater than that in patients without anemia [5.95 (2.18) vs.1.41 (1.47) cmH2O, $P<0.01$].

## Neurological imaging

At baseline, all imaging examinations were performed before undergoing LP. The brain and orbital MRI with contrast showed no imaging abnormalities that could cause intracranial hypertension or visual impairment. The common imaging features of IIH, such as empty sella (Fig 2A and 2B), DPSS (Fig 2C) and unilateral or bilateral transverse sinus stenosis (TSS, Fig 3) were found in 113 participants[73.86%]: 16 anemic, 97 non-anemic (Table 2). The most common imaging sign in both groups was empty sella (45.45% in group with anemia, 49.62% in group without anemia, respectively), followed by TSS (54.55% and 30.53%, respectively) and DPSS (22.73% and 22.14%, respectively). Compared with patients in non-anemia group, patients with anemia more likely had the sign of TSS (54.55% vs. 30.53%, $P = 0.028$). The frequency of empty sella and DPSS showed no significant differences.

## Discussion

In this single-center retrospective study, adult patients with IIH and anemia who experienced shorter disease course and had faster relief were described. IIH patients with anemia benefited

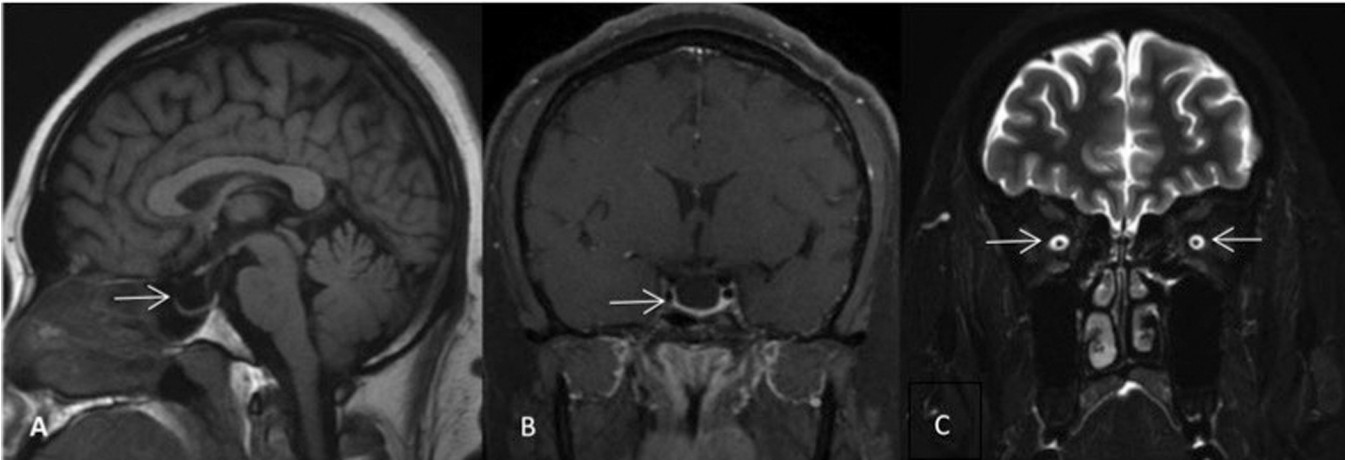

**Fig 2. Common imaging findings caused by IIH.** A and B showed empty sella on T1WI sequence; C showed DPSS in both eyes on coronal fat-depressed T2WI sequence.

from anemic treatments, and this was confirmed by the fact that there were significantly improvements in BCVA, papilledema and CSF OPs when compared to those in non-anemia group after 20 days of anemia correction and medical treatments. Moreover, patients with anemia had higher ICPs and more severe visual field damage when compared to controls.

Although the possible etiology for IIH remains unclear, obesity, rapid weight gain, anemia and obstructive sleep apnea are widely recognized as the risk factors of IIH [1–3]. This study described demographic and clinical characteristics of IIH patients with anemia, rather than the relationship between anemia and IIH.

In anemia group, 19/22 cases had IDA and 3/22 had RA. Chronic blood loss is the most common cause of IDA in these patients, especially in women suffering from menorrhagia. There were no noticeable differences in age, sex and BMI in both the groups. Most of the patients are females (86.36% in anemia group and 74.05% in control group), and this was consistent with the previous findings of occurrence of IIH more commonly in females of child bearing age [9]. It has been reported that nearly 90% of patients with IIH are obese in

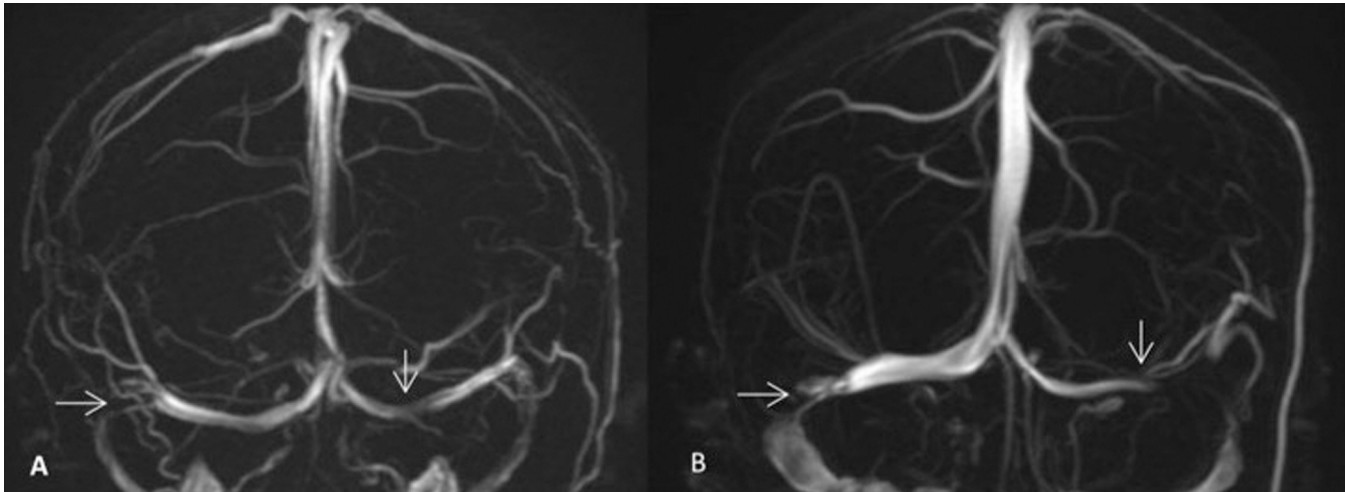

**Fig 3. TSS on MRV.** A and B showed narrowing in bilateral transverse sinus.

Caucasian population [4]. However, in this cohort, obese people accounted for less than 40% (36.36% in anemia group and 32.82% in control group), but 72.72% of cases in anemia group and 80.15% in non-anemia group were overweight (24≤BMI<28) and obese (BMI≥28). This might be because of the differences in ethnicity, racial factors and dietary structure.

In the current study, vision loss was the most common symptom in patients with IIH at presentation. At baseline (VST 0), 60/153 patients [39.22%] had severe visual deficits (BCVA<0.1), and after stratified analysis of vision loss, there were higher proportion of patients with 0.1≤BCVA<0.5 (29.41% vs 13.99%, $P$ = 0.025) and lower proportion of patients with BCVA≥0.5 (35.29% vs 61.14%, $P$ = 0.005) in the anemia group when compared to those in the control group, but no significant differences were observed between both groups in patients with severe visual impairment (BCVA<0.1, 35.29% vs 24.87%, $P$ = 0.204). Moreover, patients with anemia more likely presented pulsatile tinnitus than those in non-anemia group (54.55% vs. 29.01%, $P$ = 0.018). The mean papilledema grades were similar in both groups ($P$ = 0.082). But visual field defects in anemia group were more severe as confirmed by Humphery perimetry tests ($P$ = 0.046).

Additionally, patients with anemia had higher CSF OPs when compared to the controls. No obvious differences were found in the severity of headache and papilledema between both groups.

IIH is also referred to as "benign intracranial hypertension" [10]. However, IIH is not "benign" as it can cause extreme loss of vision. "The vision is at stake", and any delay in diagnosis or inappropriate management might lead to permanent loss of vision. The primary principle for managing IIH includes protection of vision, modification of the underlying disease and weight control. Based on re-evaluation at VST 1 after medical treatments as mentioned above, the clinical symptoms and CSF OPs of all patients showed improvement, especially in the anemia group. Firstly, the frequency of subjects with BCVA <0.1 was decreased to 5.88%, which was obviously lower than that in the control group (21.76%, $P$ = 0.031); secondly, the papilledema of eyes showed a significant improvement when compared to those in non-anemia group (0.82(0.59) vs. -0.34(0.49), $P$<0.01); thirdly, the CSF OPs reduction (cm $H_2O$) was significantly greater than those in without anemia [-5.95(2.18) vs. -1.41(1.47), $P$<0.01]. Meanwhile, Hb levels of these anemic participants were elevated, in which 5 have returned to normal. We speculated that these clinical improvements of the anemic patients in a short period of time were attributed to the reduction in ICP, and treatments for anemia played an important role during this process.

After that, follow-up at VST 2 and VST 3 showed no significant differences in visual field defects, papilledema, headache and CSF OPs between the two groups. However, the proportion of patients with BCVA≥0.5 in the anemia group was significantly higher than that in the control group (82.35% vs 65.28%, $P$ = 0.049) at VST 3, suggesting better visual prognosis in IIH patients with anemia. We speculated that the anemia treatments have brought better clinical outcomes in IIH patients with anemia, although no attempt was made to apply correction of anemia alone in this study. Several case series have reported that the symptoms of IIH might be resolved by simply correcting anemia [11,12], thus highlighting the importance of initiating an appropriate treatment for anemia. Moreover, shorter disease course is another reason for better prognosis in IIH patients with anemia, and this is because the optic nerve might not be severely damaged due to long-term increased ICP.

Various types of anemia associated with IIH have been described in some previous studies, such as IDA, hemolytic anemia, and idiopathic aplastic anemia [13,14]. Anemia induces changes in the hemodynamics, which might result in increased ICP in patients with anemia. Some case reports have discussed the relation between IDA and cerebral venous thrombosis (CVT) with IIH-like presentation [15–17]. There are some mechanisms put forwarded to

explain the association between IDA and thrombosis. Firstly, iron is considered as a regulator of thrombopoiesis by inhibiting the rise in the platelet count and consequent hypercoagulable state. Iron deficiency weakens the inhibitory function, leading to thrombocytosis and thrombosis [18]. However, the level of iron deficiency at which the switch occurs is yet to be established [19]. Another possible mechanism is that iron deficiency might induce red cell deformability and increase viscosity [20]. In our study, IDA (19/22) was considered as the most common type of anemia in the anemic group. Compared with non-anemia group, the significantly higher TSS frequency (54.55% vs. 30.53%, $P = 0.028$) in anemia group indicated the possibility of CVT based on the above mentioned mechanisms. Also the possibility of this has been taken into consideration because CVT is one of the important differential diagnoses that must be ruled out before diagnosing IIH. But the evidence of CVT was not highlighted by DSA/CTA scans. However, the presence of small vein thrombosis or not cannot be speculated. In the study conducted by Biousse *et al.*, a possible hyperviscosity mechanism that increases the venous pressure without true venous sinus thrombosis has been proposed [14]. Specific pathogenesis underlying the association of elevated ICP and anemia needs further research and confirmation.

With higher portion of narrowing transverse sinus in the anemia group, patients with anemia were more likely to present pulsatile tinnitus than those in non-anemia group (54.55% vs. 29.01%, $P = 0.018$). The association between TSS and tinnitus observed in this cohort was in line with several other clinical studies that indicated variations in intracranial fluids influenced by labyrinth hydromechanics, causing tinnitus or vertigo [21,22]. However, in light of the presence of TSS and tinnitus in IIH patients with anemia, the anemia-related hyperviscosity mentioned above might worsen the hemodynamic changes. This might also be a possible reason as to why the elevated ICP of anemia cases was significantly higher than in those without anemia ($P = 0.015$). In the study conducted by G.Chiarella et al. [23], researchers have hypothesized an alteration of venous cerebral circulation, especially TSS, might be a pathophysiological basis for IIH, even in the absence of clinical evidence of elevated ICP. But it is still under debate whether TSS is a cause of IIH or a consequence of IIH.

Neuroimaging examinations are essential for diagnosing IIH, especially the fat-suppressed Gd-enhanced MR scans of the brain and orbit are strongly recommended for evaluating the disorders involving papilledema and headaches [24]. In the present study, 113/153 (73.86%) cases with IIH had common signs of increased ICP by radiological examinations, and this was considered very helpful for accurate diagnosis of IIH. Except TSS discussed above, the rates of empty sella and DPSS showed no significant differences between anemia and non-anemia groups. Empty sella has been thought to present in the later stage of increased ICP [25]. But 45.45% of anemic cases had this sign, which indicated that empty sella could appear early during the clinical course of IIH.

The highlight of our study is that response to treatment was assessed by serially measuring the CSF OP. Visual impairment is the predominant manifestation in enrolled patients at presentation. Baseline (VST 0) BCVA was worse than the IIHTT with 75% of eyes showing BCVA 20/20 or above [26]. Relatively severe papilledema was observed in 84 (60.87%) cases with grade ≥4 on MFS. This was similar with the IIHTT Optical coherence tomography Baseline Study with baseline papilledema that was mostly moderate to severe (grade 2–4) [27], but was much more severe than several other studies [6,7,28,29]. At final follow-up VST 3, 63.04% (87/138) of eyes were with papilledema. This was in line with a prospective study conducted by Wall and George, where 64% of eyes were with papilledema at the final visit (follow-up 2–39 months) [30]. The symptoms and ICP of patients in this retrospective cohort were improved over time, implying that adequate treatment and follow-up have beneficial effects. The most important therapeutic drug in this study was acetazolamide (25mg/tablet). All patients were

treated with acetazolamide at a dose of 200 mg to 300mg/day. In clinical application, most of the participants were unable to tolerate the side effects (especially limb numbness and gastro-intestinal symptoms) induced by acetazolamide, and so large doses of drugs recommended by the guidelines [31] were not given to patients with IIH, while the treatment effects were considered acceptable, which included judging from the improvement of CSF OPs and clinical symptoms in this study.

## Limitations

However, our study had some limitations with regard to the retrospective data collection, as well as missing data when patients failed to show up to appointments due to transportation difficulties for those living outside Beijing. Besides, some patients did not complete HF perimetry tests as required, and this reduced the strength of our assessment on visual function. In addition, the power of the study was not calculated prior to starting the research, and small sample size with IIH and anemia is also one of the limitations. Long-term follow-up of this cohort is ongoing and the results will be shown in the future reports.

## Conclusions

In conclusion, the clinical characteristics of IIH patients with anemia included shorter disease duration, tendency to suffer tinnitus and higher ICP, liable to develop TSS, and better clinical visual outcomes after undergoing timely treatments of anemia-correction and decreasing ICP. These findings highlighted the importance of obtaining full blood counts in patients with IIH and initiating appropriate treatments if proved to be anemic.

## Author Contributions

**Conceptualization:** Zhonghua Ma, Jiawei Wang.

**Data curation:** Zhonghua Ma, Shilei Cui, Jingting Peng.

**Formal analysis:** Hanqiu Jiang, Chao Meng.

**Funding acquisition:** Jiawei Wang.

**Investigation:** Hanqiu Jiang, Shilei Cui.

**Methodology:** Zhonghua Ma, Hanqiu Jiang, Chao Meng, Jiawei Wang.

**Resources:** Shilei Cui, Jingting Peng.

**Supervision:** Hanqiu Jiang, Chao Meng, Jiawei Wang.

**Validation:** Jingting Peng.

**Visualization:** Chao Meng, Shilei Cui.

**Writing – original draft:** Zhonghua Ma.

**Writing – review & editing:** Zhonghua Ma, Shilei Cui, Jiawei Wang.

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
