## [Decision Letter · Decision Letter 0]

28 Apr 2020

PONE-D-20-08499

Idiopathic Intracranial Hypertension in Patients with Anemia: A Retrospective Cross-sectional Study

PLOS ONE

Dear Mr. Ma,

Thank you for submitting your manuscript to PLOS ONE. After careful consideration, we feel that it has merit but does not fully meet PLOS ONE’s publication criteria as it currently stands. Therefore, we invite you to submit a revised version of the manuscript that addresses the points raised during the review process.

The reviewers mention a several amount of issues that should be acknowledged before considering your manuscript for publication in Plos One, therefore I suggest you review each comment carefully, addressing the requested information.

We would appreciate receiving your revised manuscript by Jun 12 2020 11:59PM. To enhance the reproducibility of your results, we recommend that if applicable you deposit your laboratory protocols in protocols.io, where a protocol can be assigned its own identifier (DOI) such that it can be cited independently in the future. For instructions see: http://journals.plos.org/plosone/s/submission-guidelines#loc-laboratory-protocols

We look forward to receiving your revised manuscript.

Kind regards,

Miguel A. Barboza, MD, MSc

Academic Editor

PLOS ONE

2. In the ethics statement in the manuscript and in the online submission form, please provide additional information about the patient records used in your retrospective study, including: a) the date range (month and year) during which patients' medical records were accessed; and b) the source of the medical records analyzed in this work (e.g. hospital, institution or medical center name).

Reviewers' comments:

Reviewer's Responses to Questions

**Comments to the Author**

1. Is the manuscript technically sound, and do the data support the conclusions?

Reviewer #1: Partly

Reviewer #2: Yes

2. Has the statistical analysis been performed appropriately and rigorously? 

Reviewer #1: Yes

Reviewer #2: Yes

3. Have the authors made all data underlying the findings in their manuscript fully available?

Reviewer #1: No

Reviewer #2: Yes

4. Is the manuscript presented in an intelligible fashion and written in standard English?

Reviewer #1: Yes

Reviewer #2: Yes

5. Review Comments to the Author

Reviewer #1: This retrospective cohort study has reviewed the records of adult patients with IIH from Jan 2014 - Aug 2019 at a single center, and performed a straight-forward separation into two groups based on the presence of anemia. Based on their findings, they make the conclusion that patients with anemia have an “subacute onset, shorter disease duration, tendency to suffer tinnitus and higher ICP, liable to develop TSS, and better clinical visual outcomes after timely treatments of anemia-correction and decreasing ICP”.

In general, this report provides a satisfactory picture that there is a trend that patients with anemia tend to do better. Unfortunately, the data collected is fairly minimalistic. All improvements are lumped into a simple yes or no category, and no further follow-up information is provided on the patients and their outcomes, despite having several criteria within that category as stated in the report. There is no follow-up Hb level showing that anemia was treated or not or what they were treated with, that a follow-up opening pressure was performed, or documentation of how many have papilledema and how many have shown improvement. The long-term follow-up characterization is also quite minimal and amounts to a simple “yes or no” data field and no granularity to clinical outcome data. Minor issues also include incorrectly used terms (referring to an opening pressure as ICP, eg). Clinical outcomes has been converted into a subjective criteria without evidence of the objective criteria, except for visual outcomes.

Overall this fares closer to empirical expert opinion, and lacks several aspects of rigorous research, as best as can be expected with a retrospective cohort study.

Reviewer #2: We thank the author for submitting their paper to PLOS ONE.

This manuscript describes a longitudinal cohort study done in patients with IIH and anemia. Their major objective is to define the clinical characteristics and prognosis of patients with IIH and anemia and without it during a period of 6 years and 7 months. The main conclusion given included that patients with IIH and anemia were more prone to a sub-acute onset, shorter disease duration, tinnitus, higher intracranial pressure, transverse sinus stenosis liability and better clinical visual outcome after treatment of anemia and lowering intracranial pressure.

The authors´s article inspires a lukewarm impression and the possibility of publication would be based in an overall improvement of the manuscript. I detail a general paragraph and some specific issues (major and minor) to be addressed to improve their article.

This is an significant topic and of interest to our readers. The article seems original in the sense of specific characterization of the patients with IIH and anemia; however, the appearance of anemia (specially microcytic/ferropenic) in IIH has been described before (casuistic has defined a prevalence of up to 10%) as well as the fact that only treating anemia would decrease intensity of headaches and even stop the course of the disease in some cases.

The writing is clear; however, general organization could be improved to provide an adequate flow and make the reading more comfortable. This will also aid the author to identify the major point of view in all the sections. More attention should be placed in specific areas such as criteria (diagnosis, inclusion and exclusion), treatments and limitations. The paper could be enhanced by tables and figures related to the data depicted.

Major issues:

1. Title: The author´s annotate a tittle suggesting a retrospective cross-sectional study and later define a longitudinal cohort study. A proper analysis and definition of the type of study design is a must.

2. Materials and methods: the means of getting to the information of the patients to the database should be provided in order to validate the author´s source of research: review of data charts, radiologic tests, laboratory records, direct or indirect contact with patients, surveys, etc.

3. Criteria: This paragraph could be improved greatly.

- Diagnostic criteria should be described in the article. Even though we assume that criteria for diagnosis is homogeneous internationally, some countries could variate their way to diagnose diseases according to their data and experience.

- The course of disease was defined as acute, sub-acute and chronic. It is important to describe why and how you defined the time frame for each category, either if you chose them based on prior studies or based on your own experience. The definition for acute course needs to be added.

- Even though it is also assumed, annotating that all patients had a complete and normal neurological exam (apart from CN VI palsy if any) is needed to ensure diagnostic criteria was met.

- Campimetry is also part of the diagnostic criteria of IIH and a significant number of the patients did not have a campimetric study. Explanation of the missing information should be addressed, as well as the repercussions in diagnosis, treatment and response to treatment, as well as inherent limitations to the conclusions study.

4. Treatment and prognosis:

- This section analyzed the treatments given. A clear approach step by step according to response should be described. Whether the patients were recommended to lose weight initially, followed by or accompanied by farmacologic treatment and / or surgical measures (depending of the severity, duration or accompanying features of the headache). Current guidelines indicate weight control, acetazolamide 500 BID up to 2-4 g qd, furosemide 20-40 mg qd as adjunctive medication and surgical treatments. If the guidelines were not followed, an explanation of the reason as well as the aftermath in prognosis (response) and your overall conclusions of the study is recommended.

5. Citations: the author´s citations pertinent and current. However, some of them do not coincide with the information in the text. It is very important that they support and expand the assertions of fact not addressed by the data presented.

6. Limitations: according to the aforementioned details, this section could be greatly enriched. The repercussions and possible solutions to these limitations should also be addressed.

Minor issues and other comments:

1. Discussion: Note that abbreviations and what they stand for need to be placed before using them alone in the article. It aids in a correct understanding of the text for all our readers.

2. Figures and tables: The use of tables and / or figures may improve greatly your paper, especially for diagnostic, inclusion and exclusion criteria. A flowchart is another possibility to strengthen the readers´ understanding and focus their attention on important facts.

3. Consider modifying the term renal anemia for anemia of chronic renal disease for better and correct understanding of the term.

4. Image quality could be improved in order to enhance your article.

5. If the power of the study was calculated prior to starting the research, a description of this process is important, since it can aid in searching for an appropriate sample size and increase the chances of getting significant results. If it was not it could be added to the limitations of the study.

6. A correction and re-review is advised for further analysis and possibility of publication.

6. PLOS authors have the option to publish the peer review history of their article (what does this mean?). If published, this will include your full peer review and any attached files.

Reviewer #1: No

Reviewer #2: Yes: KARLA ALEJANDRA MORA RODRIGUEZ MD Neurology Attending Physician

---

## [Author Response · Author response to Decision Letter 0]

6 Jun 2020

Dear editors and reviewers, 

Thanks very much for your valuable comments. For more than a month, we have carefully collected the clinical data related to the article again. Our revisions and responses to these comments are as follows:

The revised part is highlighted in blue in the manuscript.

1.Please ensure that your manuscript meets PLOS ONE's style requirements, including those for file naming.

Reply: My manuscript meets PLOS ONE's style requirements.

2.In the ethics statement in the manuscript and in the online submission form, please provide additional information about the patient records used in your retrospective study, including: a) the date range (month and year) during which patients' medical records were accessed; and b) the source of the medical records analyzed in this work (e.g. hospital, institution or medical center name).

Reply: The information have been provided in “Cohort” part of the manuscript.

Reviewers’ Comments

Reviewer #1: 

1. In general, this report provides a satisfactory picture that there is a trend that patients with anemia tend to do better. Unfortunately, the data collected is fairly minimalistic. All improvements are lumped into a simple yes or no category, and no further follow-up information is provided on the patients and their outcomes, despite having several criteria within that category as stated in the report. There is no follow-up Hb level showing that anemia was treated or not or what they were treated with, that a follow-up opening pressure was performed, or documentation of how many have papilledema and how many have shown improvement. The long-term follow-up characterization is also quite minimal and amounts to a simple “yes or no” data field and no granularity to clinical outcome data. 

Reply: The information of BCVA, headache impact, papilledema grade, automated perimetry, CSF open pressure (OP) , Hb level, treatments and improvement has been added to the “Methods” and the “Results” parts in detail.

2.Minor issues also include incorrectly used terms (referring to an opening pressure as ICP, eg). Clinical outcomes has been converted into a subjective criteria without evidence of the objective criteria, except for visual outcomes.

Reply: The incorrectly used terms have been revised. Accurate and objective data of clinical outcomes has been added to the “Results” part.

Reviewer #1: 

Major issues:

1. Title: The author´s annotate a tittle suggesting a retrospective cross-sectional study and later define a longitudinal cohort study. A proper analysis and definition of the type of study design is a must.

Reply: The tittle has been revised.

2. Materials and methods: the means of getting to the information of the patients to the database should be provided in order to validate the author´s source of research: review of data charts, radiologic tests, laboratory records, direct or indirect contact with patients, surveys, etc.

Reply: The means of getting to the information of the participants have been provided in the “Materials and methods” part.

3.Criteria: This paragraph could be improved greatly.

- Diagnostic criteria should be described in the article. Even though we assume that criteria for diagnosis is homogeneous internationally, some countries could variate their way to diagnose diseases according to their data and experience.

Reply: The Dandy’s criteria has been described in detail.

- The course of disease was defined as acute, sub-acute and chronic. It is important to describe why and how you defined the time frame for each category, either if you chose them based on prior studies or based on your own experience. The definition for acute course needs to be added.

Reply: The course definition was based on our own clinical experience. The authors feel that such a definition is not suitable for use in international communication, so the definition has been deleted from the text.

- Even though it is also assumed, annotating that all patients had a complete and normal neurological exam (apart from CN VI palsy if any) is needed to ensure diagnostic criteria was met.

Reply: A normal neurological exam has been emphasized in the text.

- Campimetry is also part of the diagnostic criteria of IIH and a significant number of the patients did not have a campimetric study. Explanation of the missing information should be addressed, as well as the repercussions in diagnosis, treatment and response to treatment, as well as inherent limitations to the conclusions study.

Reply: The data of campimetry, explanation of the missing information and inherent limitations have been supplemented in the article.

4. Treatment and prognosis:

- This section analyzed the treatments given. A clear approach step by step according to response should be described. Whether the patients were recommended to lose weight initially, followed by or accompanied by farmacologic treatment and / or surgical measures (depending of the severity, duration or accompanying features of the headache). Current guidelines indicate weight control, acetazolamide 500 BID up to 2-4 g qd, furosemide 20-40 mg qd as adjunctive medication and surgical treatments. If the guidelines were not followed, an explanation of the reason as well as the aftermath in prognosis (response) and your overall conclusions of the study is recommended.

Reply: A treatment approach step by step has been described. Intolerable side effects induced by acetazolamide is the reason of why we did not follow the guideline. The explanation has been added in the text.

5.Citations: the author´s citations pertinent and current. However, some of them do not coincide with the information in the text. It is very important that they support and expand the assertions of fact not addressed by the data presented.

Reply: Citations have been rearranged.

6. Limitations: according to the aforementioned details, this section could be greatly enriched. The repercussions and possible solutions to these limitations should also be addressed.

Reply: Limitations has been enriched as recommendation.

Minor issues and other comments:

1. Discussion: Note that abbreviations and what they stand for need to be placed before using them alone in the article. It aids in a correct understanding of the text for all our readers.

Reply: Abbreviations have been rearranged as required

2. Figures and tables: The use of tables and / or figures may improve greatly your paper, especially for diagnostic, inclusion and exclusion criteria. A flowchart is another possibility to strengthen the readers´ understanding and focus their attention on important facts.

Reply: Several tables and a flowchart have been added in the manuscript.

3. Consider modifying the term renal anemia for anemia of chronic renal disease for better and correct understanding of the term.

Reply: “Renal anemia” has been defined in detail as recommended.

4. Image quality could be improved in order to enhance your article.

Reply: Images has been improved.

5. If the power of the study was calculated prior to starting the research, a description of this process is important, since it can aid in searching for an appropriate sample size and increase the chances of getting significant results. If it was not it could be added to the limitations of the study.

Reply: The power of the study was not calculated prior to starting the research. We has listed this point in the “Limitations”.

Sincerely,

Zhonghua Ma

2020-06-06

---

## [Decision Letter · Decision Letter 1]

29 Jun 2020

PONE-D-20-08499R1

Idiopathic Intracranial Hypertension in Patients with Anemia: A Retrospective Observational Study

PLOS ONE

Dear Dr. Ma,

Thank you for submitting your manuscript to PLOS ONE. After careful consideration, we feel that it has merit but does not fully meet PLOS ONE’s publication criteria as it currently stands. Therefore, we invite you to submit a revised version of the manuscript that addresses the points raised during the review process.

The manuscript improved substantially, and the authors reached all the suggestions from revisors, however, English quality should be improved in order to ease publication process, therefore, we suggest professional English editing services.

We look forward to receiving your revised manuscript.

Kind regards,

Miguel A. Barboza, MD, MSc

Academic Editor

PLOS ONE

Reviewers' comments:

Reviewer's Responses to Questions

**Comments to the Author**

1. If the authors have adequately addressed your comments raised in a previous round of review and you feel that this manuscript is now acceptable for publication, you may indicate that here to bypass the “Comments to the Author” section, enter your conflict of interest statement in the “Confidential to Editor” section, and submit your "Accept" recommendation.

Reviewer #2: All comments have been addressed

2. Is the manuscript technically sound, and do the data support the conclusions?

Reviewer #2: Yes

3. Has the statistical analysis been performed appropriately and rigorously? 

Reviewer #2: Yes

4. Have the authors made all data underlying the findings in their manuscript fully available?

Reviewer #2: Yes

5. Is the manuscript presented in an intelligible fashion and written in standard English?

Reviewer #2: Yes

6. Review Comments to the Author

Reviewer #2: The authors improved their manuscript according to suggestions made by the editing committee and reviewers. Some minor orthographic and syntactic corrections should be done prior to publication. Usually, a native English speaking aid is recommended. Overall, my opinion on the corrected version is positive.

7. PLOS authors have the option to publish the peer review history of their article (what does this mean?). If published, this will include your full peer review and any attached files.

Reviewer #2: **Yes: **KARLA A. MORA RODRIGUEZ MD NEUROLOGY

---

## [Author Response · Author response to Decision Letter 1]

11 Jul 2020

Dear editors and reviewers, 

Thanks very much for your valuable comments. Our revisions and responses to these comments are as follows:

“English quality should be improved in order to ease publication process, therefore, we suggest professional English editing services.”

Reply: English quality of this manuscript has been improved.

Reviewers’ Comments

“Some minor orthographic and syntactic corrections should be done prior to publication. Usually, a native English speaking aid is recommended.”

Reply: English quality of this manuscript has been improved.

Sincerely,

Zhonghua Ma

2020-07-06

---

## [Editor Report · Decision Letter 2]

15 Jul 2020

Idiopathic Intracranial Hypertension in Patients with Anemia: A Retrospective Observational Study

PONE-D-20-08499R2

Dear Dr. Wang,

We’re pleased to inform you that your manuscript has been judged scientifically suitable for publication and will be formally accepted for publication once it meets all outstanding technical requirements.

Kind regards,

Miguel A. Barboza, MD, MSc

Academic Editor

PLOS ONE
---

## [Editor Report · Acceptance letter]

20 Jul 2020

PONE-D-20-08499R2 

Idiopathic Intracranial Hypertension in Patients with Anemia: A Retrospective Observational Study 

Dear Dr. Wang:

I'm pleased to inform you that your manuscript has been deemed suitable for publication in PLOS ONE. Congratulations! Your manuscript is now with our production department. 

Kind regards, 

on behalf of

Dr. Miguel A. Barboza 

Academic Editor

PLOS ONE